# Characteristic Flavor Compounds and Functional Components of Fragrant Rice with Different Flavor Types

**DOI:** 10.3390/foods12112185

**Published:** 2023-05-29

**Authors:** Lin Lu, Zhanqiang Hu, Changyun Fang, Xianqiao Hu

**Affiliations:** China National Rice Research Institute, Hangzhou 310006, China; hzq8362029@163.com (Z.H.); fcyst88@163.com (C.F.); hxhxqia@aliyun.com (X.H.)

**Keywords:** popcorn flavor, corn flavor, lotus root flavor, fragrant rice, GC-MS, GC-O

## Abstract

Fragrant rice has various flavor types, mainly the popcorn flavor, corn flavor and lotus root flavor. Chinese fragrant rice from China and Thai fragrant rice from Thailand were analyzed. GC-MS was used to determine the volatile compounds of fragrant rice. It was found that there were 28 identical volatile compounds between Chinese and Thai fragrant rice. The key compounds of different flavor types of fragrant rice were obtained by comparing the common volatile compounds. The key compounds of the popcorn flavor were 2-butyl-2-octenal, 4-methylbenzaldehyde, ethyl 4-(ethyloxy)-2-oxobut-3-enoate and methoxy-phenyl-oxime. The key compounds of the corn flavor were 2,2′,5,5′-tetramethyl-1,1′-biphenyl, 1-hexadecanol, 5-ethylcyclopent-1-enecarboxaldehyde and cis-muurola-4(14), 5-diene. By using a combination of GC-MS and GC-O, the flavor spectrogram of fragrant rice was constructed, and the characteristic flavor compounds of each flavor type were identified. It was found that the characteristic flavor compounds of the popcorn flavor were 2-butyl-2-octenal, 2-pentadecanone, 2-acetyl-1-pyrroline, 4-methylbenzaldehyde, 6,10,14-trimethyl-2-pentadecanone, phenol and methoxy-phenyl-oxime. The characteristic flavor compounds of the corn flavor were 1-octen-3-ol, 2-acetyl-1-pyrroline, 3-methylbutyl 2-ethylhexanoate, methylcarbamate, phenol, nonanal and cis-muurola-4(14), 5-diene. The characteristic flavor compounds of the lotus root flavor were 2-acetyl-1-pyrroline, 10-undecenal, 1-nonanol, 1-undecanol, phytol and 6,10,14-trimethyl-2-pentadecanone. The resistant starch content of lotus root flavor rice was relatively high (0.8%). The correlation between flavor volatiles and functional components was analyzed. It was found that the fat acidity of fragrant rice was highly correlated (R = 0.86) with the characteristic flavor compounds, such as 1-octen-3-ol, 2-butyl-2-octenal and 3-methylbutyl-2-ethylhexanoate. The characteristic flavor compounds had an interactive contribution to the production of the different flavor types of fragrant rice.

## 1. Introduction

Rice is one of the main food crops around the world. The demand for rice has risen to a higher level, and the edible quality of rice has become of great concern. The flavor of rice is one of the most important criteria for consumers in selecting rice [1,2]. Consumers prefer rice with a pleasant fragrance and flavor. A slight change in the flavor greatly affects the consumer’s purchase intention. Therefore, the evaluation of and research on rice flavor plays an important role in promoting rice market competition and the agricultural economy. The volatile aroma compounds in rice play a decisive role in its flavor, and they have received much attention. For half a century, the effects of the planting conditions, harvesting conditions, storage conditions, genes and other factors on the flavor of rice have been studied, as well as the characteristic compounds and formation mechanisms. At present, the evaluation of fragrant rice and the identification of characteristic volatile compounds are indispensable.

It has been reported that more than 300 types of volatile components can be detected in rice. The flavor of rice consists of a complex mixture of various volatile active compounds. It has been confirmed that there are a large number of compounds and several main aroma compounds in the flavor of rice [3,4]. It has been reported that the main volatile components of fragrant rice and non-fragrant rice are 2-acetyl-1-pyrroline, isoamyl alcohol, 1-octen-3-ol, 2-octenal, 2-ethyl-1-hexanol, etc. [5]. In particular, 2-acetyl-1-pyrroline (2-AP) is considered to be a characteristic volatile component that displays not only the fragrance of popcorn but also the fragrance of leaves or flowers [6]. The aroma intensity of rice mainly depends on the 2-AP content, which varies among rice varieties [7]. It was found that the sensory evaluation results of rice varieties that all contained 2-AP were quite different [2,8]. The flavor quality of rice is not only determined by 2-AP, but is the comprehensive interaction of countless volatile compounds. Five sulfur volatiles, namely dimethyl sulfide, 3-methyl-2-butene-1-thiol, 2-methyl-3-furanthiol, dimethyl trisulfide and methional, were found to display aroma activity in fragrant rice [9]. It was reported that ethyl acetate, etheyl octanoat and other esters could influence the flavor of rice, making the aroma stronger. Some organic heterocyclic compounds, such as pyrazine, pyridine and furan, were closely related to the sweetness of rice. In addition, many other volatile compounds in rice contributed to the aroma, such as indole, p-xylene, 2-acetyl-2-thiazoline, etc. [10].

Sensory evaluation is traditionally adopted to assess the flavor and aroma of rice. It is difficult to unify standards for evaluators due to their different preferences, as well as individual errors. The fatigue caused by subjective examination greatly affects the accuracy of sensory evaluation. Gas chromatography–mass spectrometry (GC-MS) has unique advantages in the analysis of volatile compounds and has been effectively applied to the determination of volatile compounds in rice [9,11]. Studies of the sensory description of rice flavor have been carried out widely [12,13]. Gas chromatography–olfactometry (GC-O) is an effective technology to confirm the volatile compounds that play a key role in the overall flavor. It is frequently used to identify characteristic volatile compounds and determine the flavor intensity of food, and it can be used to determine the characteristic compounds of rice flavor.

Resistant starch and fat acidity are functional components of rice and also exist in fragrant rice. Resistant starch cannot be digested in the human small intestine but can be fermented and decomposed by the microbial flora in the colon. It has physiological functions, including stabilizing postprandial blood sugar, controlling diabetes, reducing the blood cholesterol content and reducing the incidence rate of obesity and stones [14,15]. It has been proven that rice with a high level of resistant starch has a low glycemic index. Fatty acid is accumulated in rice due to the difficulty of its use by microorganisms, making its content gradually increase during storage, further producing an unpleasant odor [16]. The change in fat acidity during rice storage is closely related to its flavor composition [17]. However, the relationship between the functional components of fragrant rice and its flavor volatile compounds needs to be studied.

In this study, the contributions of the volatile compounds of fragrant rice to the formation of the flavor type, and the relationships between the functional components and volatile compounds, were studied.

## 2. Materials and Methods

### 2.1. Experimental Materials

The experimental materials were eight fragrant rice samples, including Thai fragrant rice (from Thailand) and seven types of Chinese fragrant rice (from China). The Thai fragrant rice was numbered SY1. The varieties of Chinese fragrant rice were GouDanDang, JinGaoNen, TongHe, HongHe, GouJinGao, GouDang 1 and BaiXiangHe, which were numbered SY2, SY3, SY4, SY5, SY6, SY7 and SY8, respectively. All samples were stored in an environment with 25% relative humidity and a temperature of 20 °C.

### 2.2. Determination by GC-MS

First, 2.0 g of milled rice was weighed into a 15 mL glass extraction bottle. The water/rice ratio of rice samples was 1.2:1. After soaking for 30 min, the bottle with the rice sample was steamed for 40 min and then braised for 20 min. Then, it was inserted into an SPME extraction head and placed in a water bath at 80 °C for 30 min in order to enrich the volatile flavor compounds. The extraction head was inserted into the gas chromatography injection port for GC-QTOF-MS analysis using manual injection mode.

The parameters of GC-QTOF-MS were as follows: the chromatographic column was the DB-WAX (30 m × 0.25 mm × 0.25 μm, Agilent Technologies Co., Santa Clara, CA, USA), the column temperature was maintained at 40 °C for 5 min and increased to 230 °C at the speed of 7 °C/min; the temperature of the sample inlet was 250 °C, the carrier gas was high-purity helium (purity > 99.999%), the flow rate was 1.5 mL/min, the pressure was constant, and there was no split flow; the temperature of the EI ion source was 230 °C, and the ion source voltage was 70 eV; full scan mode was used, and the scanning range was *m*/*z* 35–400. All samples were assessed in triplicate.

After determination, Agilent Masshunter Qualitative Analysis was used to retrieve the compounds. Then, results were imported into Mass Profiler Professional for screening according to the frequency of occurrence, and finally retrieved using the NIST 14 spectrogram library for compound identification. The content of the volatile compounds was expressed as the relative content, i.e., the percentage of its peak area within the total peak area.

### 2.3. Determination by GC-O and Evaluation

In this study, GC-O was performed using the combination of a gas chromatograph (Agilent 6890 Series, Agilent Technologies Co.) and an olfactometer (Brechbuhler 9100 Series, Brechbuhler AG, Uzville, Switzerland). The sample treatment, sample injection method and gas chromatographic conditions were the same as those used in Section 2.2 for GC-QTOF, and the temperature of the olfactometer transmission line was 240 °C.

Four sensory evaluators who had received professional fragrance training and aroma sensory training were selected to describe the rice flavor. During the smelling period of GC-O, the sensory evaluator recorded the retention time, aroma description and odor intensity of the compound. The scores for odor intensity were 1 (very weak), 2 (weak), 3 (average), 4 (strong) and 5 (very strong). The times and the average intensity scores of each flavor compound smelt by the evaluators were calculated.

### 2.4. Identification of Flavor Type of Fragrant Rice

All rice samples were identified according to Chinese agricultural industry standard NY/T 596-2002. Accurately 2.0 g of rice was placed into a brown glass bottle with 50 mL of distilled water. It was covered and placed in a boiling water bath for 15 min, and then removed and cooled slightly. The evaluator shook the glass bottle gently before evaluation, and then opened it and sniffed the steam. All evaluators reached an agreement on the flavor type of each rice sample.

In this experiment, 8 rice samples were identified for the different flavor types: SY1, SY6, SY7 and SY8 had a popcorn flavor; SY2, SY3 and SY4 had a corn flavor; SY5 had a lotus root flavor.

### 2.5. Determination of Functional Components

Resistant starch was measured according to the method of Zhou et al. (2022) [18]. Fat acidity was measured according to the method of Jiang et al. (2020) [19].

### 2.6. Neural Network Model

A back-propagation (BP) neural network was trained with the volatile compounds as the input and the functional components as the output. The Tansig and Purelin functions were selected as the activation functions of the hidden layer and the transfer functions of the output layer of the neural network. The number of neurons in the hidden layer of this network was set to 4 because of the fast convergence speed and small mean squared error.

## 3. Results and Discussion

### 3.1. Analysis of Common Volatile Compounds of Fragrant Rice

A total of 188 volatile flavor compounds were detected in all fragrant rice samples by GC-MS, including 26 alcohols, 3 phenols, 22 aldehydes, 6 acids, 25 esters, 66 hydrocarbons, 23 ketones and 17 heterocycles. The number of compounds and the total content of each substance in the fragrant rice were different, as seen in Table 1. In all fragrant rice samples, there were seven compounds that were highly prevalent: 1-octen-3-ol, 3,4-dihydroxyphenylglycol, 4-(1-methylpropyl)phenol, nonanal, trans-2-octenal, geranylaceton and 2-acetyl-1-pyrroline. The content of nonanal was the largest (7.70~15.44%). According to literature reports [20], nonanal has a strong fragrance of fat, and it has the fragrance of orange and rose when it is diluted. Meanwhile, 2-AP mainly shows a typical popcorn fragrance, and 1-octen-3-ol has an earthy fragrance, delicate fragrance and greasy and fungal fragrance. Geranylaceton has a floral fragrance, leaf fragrance and fruity fragrance.

### 3.2. Comparison of Chinese and Thai Fragrant Rice

Popcorn flavor samples (SY6, SY7, SY8) of Chinese fragrant rice were compared with Thai fragrant rice (SY1). According to the results of GC-QTOF-MS, 56 volatile flavor compounds were detected in Thai fragrant rice, while the volatile compounds of SY6, SY7 and SY8 were less numerous than those of Thai fragrant rice, with 44, 38 and 48, respectively. As seen in Table 2, there were 28 identical volatile compounds in three samples of Chinese fragrant rice and Thai fragrant rice, including three alcohols, two phenols, seven aldehydes, two acids, three hydrocarbons, three ketones, two esters and six heterocycles. It was speculated that these volatile compounds were the main factors forming the flavor of popcorn. However, the content of these 28 volatile compounds varied. As for alcohols, the content of 1-octen-3-ol and cyclooctyl alcohol in the Chinese fragrant rice was higher than that in Thai fragrant rice, which was consistent with the conclusions of Zhang and Yuan (2014) [21] on the flavor of fragrant rice. Meanwhile, 4-(1-methylpropyl)-phenol was detected in all samples, and its content in Chinese fragrant rice was higher than that in Thai fragrant rice. This result differed from the phenolic compounds found in Khao Dawk Mali 105 fragrant rice analyzed by Mahatheeranont et al. (2001) [22], which was due to the different flavor types of fragrant rice used.

Aldehydes have a lower threshold value and have a greater impact on the flavor. Nonanal was detected in all four samples, and its content was the largest; it was larger in Chinese fragrant rice than in Thai fragrant rice. The content of 4-methyl-3-cyclohexene-1-carboxaldehyde and 4-methylbenzaldehyde was relatively low. There were only two types of acids, which were both fatty acids. These fatty acids might be derived from the hydrolysis of lipids, in which the content of long-chain fatty acids was high, but it had little impact on the overall flavor because of its large molecular weight and low volatility. The content of 3,3-dimethylhexane in Thai fragrant rice was relatively high, but it had little contribution to the popcorn flavor due to the minor influence of hydrocarbons on the flavor. The total content of ketones in Thai fragrant rice was higher than that in Chinese fragrant rice, while the esters showed the opposite tendency. The total content of heterocyclic compounds was relatively high, for which all fragrant rice samples exceeded 28%, except for SY8. Moreover, the content of 2-pentyl furan was the largest, reaching 17.20~33.53%; it has a fruity, bean-like or earthy fragrance.

### 3.3. Comparison of Different Flavor Types of Fragrant Rice

The volatile flavor compounds of fragrant rice with a popcorn flavor, corn flavor and lotus root flavor were compared. The compounds detected in all fragrant rice samples of the same flavor type were common compounds of the flavor type. Then, the coincidence of common compounds among the three flavor types was analyzed, and the non-overlapping compounds were regarded as the key compounds of each flavor type.

There were 13 and 20 common compounds in the popcorn flavor and corn flavor, respectively. As seen in Figure 1, there were nine common compounds between the popcorn and corn flavors, and 14 common compounds between the corn and lotus root flavors. Moreover, seven common compounds were detected in all three flavor types, and 2-pentylthiophene and 3,3-dimethylhexane, which were detected in the popcorn and corn flavors, did not appear in the compounds of the lotus root flavor. By comparison of the common compounds of the three flavor types, the key compounds of the popcorn flavor were identified as 2-butyl-2-octenal, 4-methylbenzaldehyde, ethyl 4-(ethyloxy)-2-oxobut-3-enoate and methoxy-phenyl-oxime. The key compounds of the corn flavor were 2,2′,5,5′-tetramethyl-1,1′-biphenyl, 1-hexadecanol, 5-ethylcyclopent-1-enecarboxaldehyde and cis-muurola-4(14), 5-diene.

### 3.4. Construction of Flavor Spectrogram of Fragrant Rice

The results of GC-MS only reveal the volatile compounds of fragrant rice, and they cannot identify the characteristic compounds that play an important role in the overall flavor. The influence of a certain compound on the overall flavor is closely related to the threshold value of the compound, in addition to its content. GC-O was used for the construction of a flavor spectrogram for the fragrant rice.

The flavor spectrogram of fragrant rice was constructed using the sensory smelling results regarding the volatile compounds of fragrant rice, obtained via GC-O. It was composed of the identification frequency (FQ) and average intensity (AI) of the volatile compounds recognized by the sensory evaluators, as shown in Figure 2. The identification frequency was defined as the number of evaluators who could identify a certain flavor among the four sensory evaluators participating in the olfaction test. Specifically, 0 ≤ FQ ≤ 4, and the greater the value, the more evaluators could recognize the compound—in other words, the greater the contribution to the overall flavor. The average intensity was defined as the average of the sensory evaluator’s score of the odor intensity. Moreover, 0 ≤ AI ≤ 5.0, and the higher the score, the higher the flavor intensity of the compound, the more obvious the fragrance and the greater the influence on the formation of the flavor type.

### 3.5. Identification of Characteristic Flavor Compounds of Fragrant Rice

In order to further identify the characteristic compounds of fragrant rice with different flavor types, the combination of GC-MS and GC-O was used. According to the retention time of GC-MS and the smelling time of the sensory evaluators, 62 volatile flavor compounds were identified in all fragrant rice samples by at least one evaluator. There were 19 flavor compounds detected by the sensory evaluators in two or more types of fragrant rice. Their retention times, compound names and flavor descriptions are listed in Table 3. In the same flavor type, the compound with the highest identification frequency (FQ ≥ 3) or highest average intensity (AI ≥ 3.0) in two or more samples was regarded as the characteristic flavor compound of this flavor type. There were seven characteristic flavor compounds in popcorn flavor rice, namely 2-butyl-2-octenal, 2-pentadecanone, 2-acetyl-1-pyrroline, 4-methylbenzaldehyde, 6,10,14-trimethyl-2-pentadecanone, phenol and methoxy-phenyl-oxime, of which three compounds were also key compounds, as seen in Section 3.3. In corn flavor rice, there were also seven characteristic flavor substances, namely 1-octen-3-ol, 2-acetyl-1-pyrroline, 3-methylbutyl-2-ethylhexanoate, methylcarbamate, phenol, nonanal and cis-muurola-4(14), 5-diene, and one of them was the key compound. In lotus root flavor rice, there were six characteristic flavor substances, namely 2-acetyl-1-pyrroline, 10-undecenal, 1-nonanol, 1-undecanol, phytol and 6,10,14-trimethyl-2-pentadecanone. These characteristic flavor compounds had a great contribution to the production of the flavor type of fragrant rice.

Among all characteristic flavor compounds, 2-AP existed in all fragrant rice samples with the highest identification frequency (FQ = 4) and the highest average intensity (AI = 4.0~4.8). Therefore, 2-AP contributed to all flavor types and was the main component of fragrant rice, which is consistent with previous reports on the flavor of fragrant rice. Buttery et al. (1982) [23] pointed out that 2-AP was the main component of the aroma of fragrant rice. This discovery is regarded by most scholars as a milestone achievement in the study of the aroma of fragrant rice. In a subsequent study, the researchers obtained consistent conclusions from different varieties of fragrant rice. It was reported that 2-AP had a very low threshold of 0.1 μg/kg [10], so it could still show a strong flavor of popcorn at low content. Organic cultivation produced fragrant rice with lower yields that had higher 2-AP content [24]. Different from previous studies, this study identified the characteristic flavor compounds of each flavor type of fragrant rice. The flavor type of fragrant rice could not be represented by the single characteristic flavor compounds, but was the result of the mixing and interactions of various characteristic flavor compounds.

### 3.6. Correlation Analysis between Flavor Volatiles and Functional Components

The resistant starch content of the fragrant rice was 0.3~0.8%, and the content of SY4 and SY5 was relatively high. The resistant starch content of the lotus root flavor rice (0.8%) was slightly higher than that of the corn flavor (0.6~0.7%), and the popcorn flavor had the lowest (0.3~0.5%). The content of resistant starch had no significant effect on the different flavor types of fragrant rice. The resistant starch content of the different flavor types varied, and that of lotus root flavor rice was relatively high (0.8%). The fat acidity of fragrant rice was 15.5~22.8 mg/100 g, and that of the popcorn flavor rice was relatively larger. The higher the fat acidity, the weaker the popcorn flavor. According to the correlation analysis results, the single correlation between fat acidity and volatile compounds was low, but the integrated correlation with aldehydes, esters and alcohols was high; this differed from the single correlation results of previous studies. It was found that the fat acidity of fragrant rice was associated with the synergy of flavor volatile compounds, which could affect the strength of the original flavor. Through the identification of the BPNN model, the fat acidity of fragrant rice was highly correlated (R = 0.86) with the characteristic flavor compounds, such as 1-octen-3-ol, 2-butyl-2-octenal and 3-methylbutyl-2-ethylhexanoate, which is partially consistent with the results of previous studies [25,26,27]. It was indicated that these compounds were the main flavor compounds that changed the flavor type of the fragrant rice during storage.

## 4. Conclusions

A total of 188 volatile compounds were detected in Thai fragrant rice and Chinese fragrant rice by GC-MS, including 26 alcohols, 3 phenols, 22 aldehydes, 6 acids, 25 esters, 66 hydrocarbons, 23 ketones and 17 heterocycles. The types and content of volatile compounds in fragrant rice with different flavor types were different. There were 28 identical volatile compounds between Chinese fragrant rice and Thai fragrant rice. The key compounds of different flavor types of fragrant rice were obtained by comparing the common volatile compounds. The key compounds of the popcorn flavor were 2-butyl-2-octenal, 4-methylbenzaldehyde, ethyl 4-(ethyloxy)-2-oxobut-3-enoate and methoxy-phenyl-oxime. The key compounds of the corn flavor were 2,2′,5,5′-tetramethyl-1,1′-biphenyl, 1-hexadecanol, 5-ethylcyclopent-1-enecarboxaldehyde and cis-muurola-4(14), 5-diene. By using a combination of GC-MS and GC-O, the flavor spectrogram of fragrant rice was constructed, and the characteristic flavor compounds of each flavor type were identified. The characteristic flavor compounds of the popcorn flavor were 2-butyl-2-octenal, 2-pentadecanone, 2-acetyl-1-pyrroline, 4-methylbenzaldehyde, 6,10,14-trimethyl-2-pentadecanone, phenol and methoxy-phenyl-oxime. The characteristic flavor compounds of the corn flavor were 1-octen-3-ol, 2-acetyl-1-pyrroline, 3-methylbutyl 2-ethylhexanoate, methylcarbamate, phenol, nonanal and cis-muurola-4(14), 5-diene. The characteristic flavor compounds of the lotus root flavor were 2-acetyl-1-pyrroline, 10-undecenal, 1-nonanol, 1-undecanol, phytol and 6,10,14-trimethyl-2-pentadecanone. The resistant starch content of the different flavor types varied, and that of the lotus root flavor rice was relatively high. The fat acidity of the fragrant rice was highly correlated with the characteristic flavor compounds, such as 1-octen-3-ol, 2-butyl-2-octenal and 3-methylbutyl-2-ethylhexanoate. These characteristic flavor compounds had an interactive contribution to the production of the different flavor types of fragrant rice.

## Figures and Tables

**Figure 1 foods-12-02185-f001:**
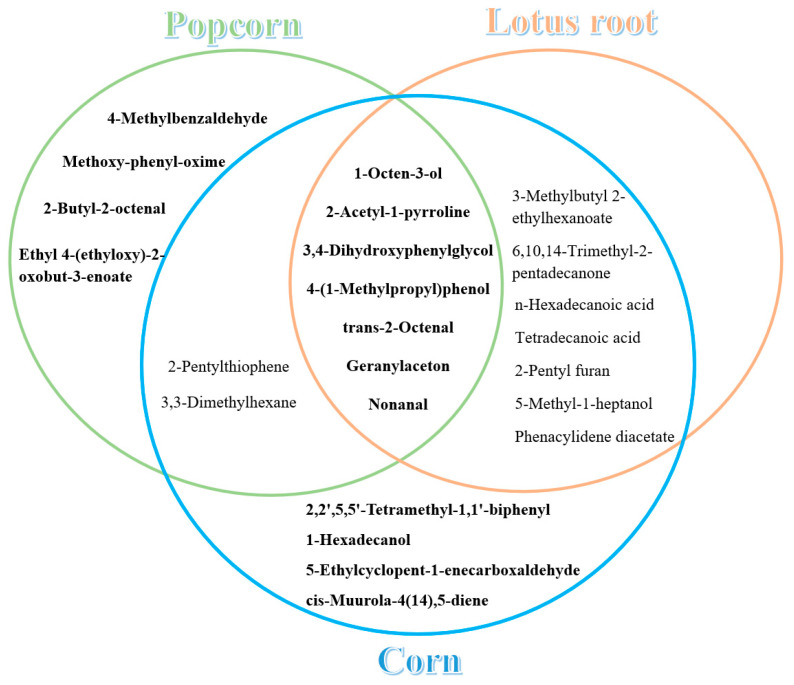
Coincidence diagram of common compounds of fragrant rice with three flavor types.

**Figure 2 foods-12-02185-f002:**
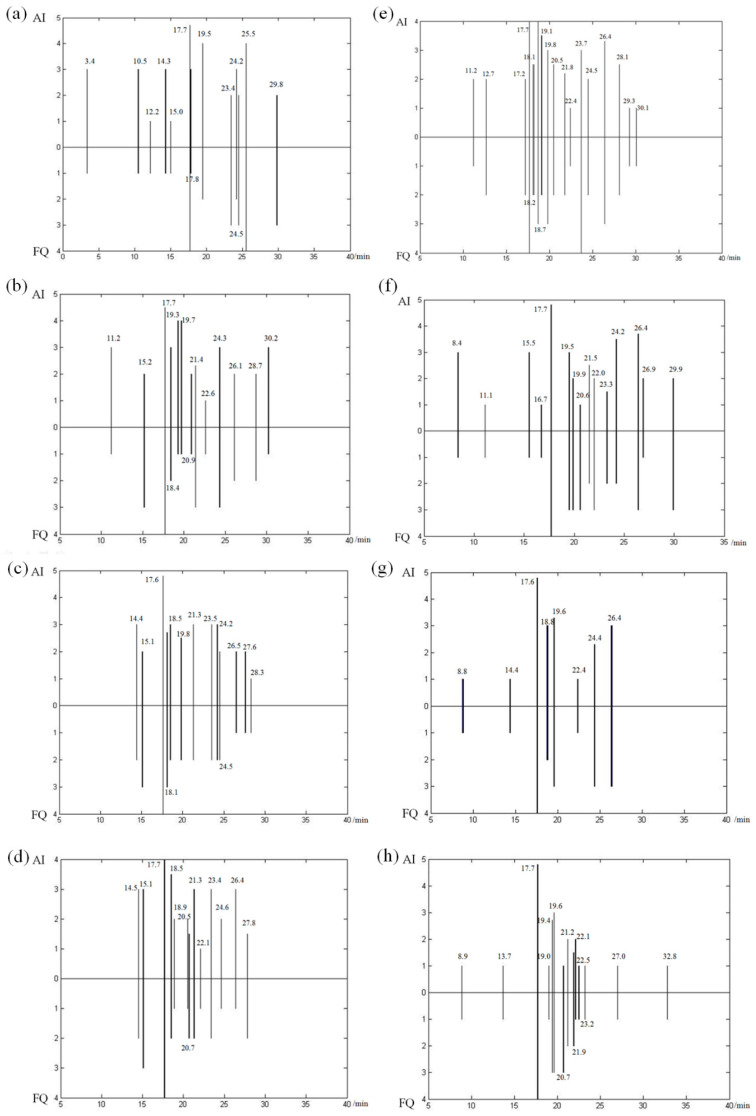
Flavor spectrogram of fragrant rice ((**a**)-SY1; (**b**)-SY2; (**c**)-SY3; (**d**)-SY4; (**e**)-SY5; (**f**)-SY6; (**g**)-SY7; (**h**)-SY8).

**Table 1 foods-12-02185-t001:** The number of compounds and the total content of each substance in fragrant rice.

Sample Number	Item	Alcohols	Phenols	Aldehydes	Acids
SY1	Number of compounds	7	2	11	2
	Total content %	11.79	1.21	26.79	2.35
SY2	Number of compounds	13	2	9	2
	Total content %	26.66	1.70	26.36	5.11
SY3	Number of compounds	9	2	7	3
	Total content %	14.31	1.17	23.31	16.88
SY4	Number of compounds	4	1	4	2
	Total content %	6.37	0.55	18.02	3.20
SY5	Number of compounds	14	2	12	2
	Total content %	14.94	9.47	23.44	9.50
SY6	Number of compounds	6	2	7	1
	Total content %	10.01	1.45	27.17	0.99
SY7	Number of compounds	4	1	8	2
	Total content %	7.81	1.08	26.55	1.29
SY8	Number of compounds	6	2	9	3
	Total content %	12.44	5.81	38.09	1.78
**Sample Number**	**Item**	**Hydrocarbons**	**Ketones**	**Esters**	**Heterocycles**
SY1	Number of compounds	11	6	8	9
	Total content %	14.03	8.85	5.47	29.50
SY2	Number of compounds	20	4	7	5
	Total content %	14.75	3.17	4.65	17.62
SY3	Number of compounds	23	4	10	8
	Total content %	12.64	3.86	8.51	19.32
SY4	Number of compounds	15	6	10	7
	Total content %	20.27	13.81	11.79	25.99
SY5	Number of compounds	22	8	11	7
	Total content %	12.93	13.85	5.01	10.85
SY6	Number of compounds	11	9	3	5
	Total content %	11.47	18.40	2.11	28.40
SY7	Number of compounds	7	5	5	6
	Total content %	9.78	2.65	7.80	43.04
SY8	Number of compounds	10	12	2	4
	Total content %	7.13	26.88	1.49	6.38

**Table 2 foods-12-02185-t002:** Identical volatile compounds in Chinese fragrant rice and Thai fragrant rice.

Volatile Compounds	Content (%)			
	SY1	SY6	SY7	SY8
Alcohols				
1-Octen-3-ol	2.23	3.69	4.13	4.14
3,4-Dihydroxyphenylglycol	2.45	2.19	1.40	0.92
Cyclooctyl alcohol	2.31	—	—	5.11
Phenols				
Phenol	0.26	0.37	—	—
4-(1-Methylpropyl)phenol	0.95	1.08	1.08	1.20
Aldehydes				
2-Butyl-2-octenal	2.14	4.33	2.73	—
4-Methyl-3-cyclohexene-1-carboxaldehyde	0.71	1.62	—	2.16
5-Ethylcyclopent-1-enecarboxaldehyde	1.01	—	1.60	—
Benzaldehyde	4.85	—	—	5.09
4-Methylbenzaldehyde	0.46	3.28	2.09	3.52
Nonanal	7.70	9.40	10.00	11.07
trans-2-Octenal	2.70	3.28	2.70	4.80
Acids				
n-Hexadecanoic acid	1.99	—	0.89	1.09
Tetradecanoic acid	0.36	—	0.40	—
Hydrocarbons				
2-Decenal	0.71	0.50	—	—
3,3-Dimethylhexane	3.20	0.62	1.15	0.70
m-Aminophenylacetylene	0.31	—	0.87	0.66
Ketones				
2-Pentadecanone	1.59	—	0.42	—
2-(Formyloxy)-1-phenylethanone	3.37	3.74	—	—
Geranylaceton	2.88	0.58	0.60	1.01
Heterocycles				
2-(p-Fluorophenyl)-1-methylbenzimidazole	1.70	1.24	—	—
2-Acetyl-1-pyrroline	0.68	2.00	1.63	1.98
2-n-Butyl furan	0.65	—	1.60	1.75
2-Pentyl furan	17.20	22.07	33.53	—
Methoxy-phenyl-oxime	6.97	2.57	4.19	1.99
2-Pentylthiophene	1.18	0.51	1.10	0.65
Esters				
Ethyl 4-(ethyloxy)-2-oxobut-3-enoate	0.38	0.81	0.68	0.80
Methyl p-(2-phenyl-1-benzimidazolyl)benzoate	0.34	0.63	0.48	—

**Table 3 foods-12-02185-t003:** Retention time, compound names and flavor descriptions of volatile compounds.

Retention Time (min)	Compound Name	Flavor Description
8.9	2-n-Butyl furan	Sweet/floral fragrance
11.2	2-Pentyl furan	Sweet/grass/fruity fragrance
14.5	Nonanal	Floral fragrance/a smell of burnt oil
15.1	1-Octen-3-ol	A smell of earth/grass/sulfur
17.6	2-Acetyl-1-pyrroline	Popcorn fragrance
18.4	cis-Muurola-4(14), 5-diene	Corn fragrance
19.6	2-Butyl-2-octenal	Corn/sweet/floral fragrance
20.7	Methoxy-phenyl-oxime	Floral fragrance
21.3	3-Methylbutyl 2-ethylhexanoate	Roasted/smelly/woody, slightly pungent
22.1	4-(1-Methylpropyl)phenol	Popcorn/corn/cooked rice fragrance
22.6	Geranylaceton	Floral/fresh/fruity fragrance
23.5	Methylcarbamate	Cooked rice fragrance/roasted
24.2	Phenol	Corn fragrance, the smell of earth/grass
24.4	2-Pentadecanone	Sweet fragrance/greasy
26.4	6,10,14-Trimethyl-2-pentadecanone	Creamy/corn/floral fragrance
27.0	4-Hydroxy-2-methylacetophenone	Mango/creamy fragrance
27.6	2,2′,5,5′-Tetramethyl-1,1′-biphenyl	Roasted/burned
29.8	4-Methylbenzaldehyde	Sweet/floral fragrance, slightly sour
30.1	m-Aminophenylacetylene	Floral/creamy fragrance

## Data Availability

The datasets generated for this study are available on request to the corresponding author.

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
