# Peer review of "Characteristic Flavor Compounds and Functional Components of Fragrant Rice with Different Flavor Types"

_foods, 2023, doi:10.3390/foods12112185_

Round 1

Reviewer 1 Report

The manuscript presents the volatile compounds and functional components of fragrant rice from Thailand and China. This topic is interesting because the rice characteristic aroma is a complex system; however, there is a lot of information in this sense.

I recommend that the authors should include more specific information (factors affecting rice aroma) about the experimental materials of this work research such as proximal composition, storage period, storage conditions, planting factors, and genetic factors.

I recommend that the authors should include an electronic nose analysis to evaluate the aroma quality of rice samples, too.

The authors have a specific reason to compare the Chinese and Thailand fragrant rice.  

I recommend that the authors should include the most current references on this subject because they are few and old. 

Author Response

I recommend that the authors should include more specific information (factors affecting rice aroma) about the experimental materials of this work research such as proximal composition, storage period, storage conditions, planting factors, and genetic factors.

Response 1: Some information such as storage conditions have been added. And, planting factors and genetic factors can be considered for further study with the increase in the number of varieties.

I recommend that the authors should include an electronic nose analysis to evaluate the aroma quality of rice samples, too. The authors have a specific reason to compare the Chinese and Thailand fragrant rice.  

Response 2:  Chinese and Thailand fragrant rice are different in volatile compound, which can be confirmed by the traditional sensory evaluation. Besides, there were many previous studies on the electronic nose analysis for rice, which focused on the discrimination of rice samples. In this study, the specific volatile compounds of Chinese and Thailand fragrant rice were our concentration.

I recommend that the authors should include the most current references on this subject because they are few and old. 

Response 3:  Some references have been replaced with the current ones.

Reviewer 2 Report

Specific points:

In this study, the authors presented a wide range of volatile compounds as well as the content of individual the functional components of fragrant rice. This kind of analysis can help in understanding the reasons for acceptability/unacceptability of certain varieties of fragrant rice.

 L: 10-11: „The key compounds of different flavor types...“– Please specify which are the key compounds.

 L: 12: „Popcorn flavor and corn flavor rice had both four key compounds.“ - Please specify which are the 4 key compounds.

 L: 12-14: „By using the combination of GC-12 MS and GC-O, the flavor spectrogram of fragrant rice was constructed, and the characteristic flavor compounds of each flavor type were identified.“ - I think that instead of this sentence, the most significant registered components for each type of flavor should be specified.

 L:14-15:„ It was resulted that there were 7, 7 and 6 characteristic flavor compounds in popcorn, corn and lotus root flavor respectively.“ - I believe that it should be specifically stated which components these are.

 L:16-17: „ The 16 correlation between flavor volatiles and functional components was analyzed.“- I believe that the result of the correlation analysis should be stated.

 L:17-18: „ The resistant starch  content of lotus root flavor rice was relatively high.” - Please provide specific values.

L: 18-20: “Fat acidity of fragrant rice was highly correlated with the characteristic flavor compounds…” - Please state the value of the correlation coefficient.

The abstract lacks a conclusion.

In general, I think that it is necessary to reformulate the abstract and specify the concrete results of the analysis. I find the abstract in this form to be very general/non-specific.

Keywords: Keywords should complement and not repeat the title. The aim of keywords is to increase visibility of article in the databases, therefore more general words might be relevant, or words that describe the work, but are not included in the title. So keywords need to be changed (the ones in the title should not be repeated).

L: 24: „Rice is the main food crop in the world.“ - Rather, I would say - one of the main...

 Figure 1. Congratulations to the authors on a very clear and picturesque diagram :)

 L:264-265 „The resistant starch content of lotus root flavor rice was slightly higher than that of the corn flavor, and the popcorn flavor was the lowest.” - I think this is a general sentence. Please give specific values.

 L:267-268 „The resistant starch content of different flavor type was different, and that of lotus  root flavor rice was relatively high.“ - Same comment as above. Please give specific values.

 L.262:   3.6. Correlation analysis between flavor volatiles and functional components - Please provide specific correlation coefficients/data for the relationships you state. You can display them in this part of the text or in a table.

 __________________

All my suggestions are for improving the manuscript. I hope all the suggestions are clear.

Best regards.

Author Response

 L: 10-11: „The key compounds of different flavor types...“– Please specify which are the key compounds.

Response 1: The key compounds have been specified.

 L: 12: „Popcorn flavor and corn flavor rice had both four key compounds.“ - Please specify which are the 4 key compounds.

Response 2: The 4 key compounds have been specified.

 L: 12-14: „By using the combination of GC-12 MS and GC-O, the flavor spectrogram of fragrant rice was constructed, and the characteristic flavor compounds of each flavor type were identified.“ - I think that instead of this sentence, the most significant registered components for each type of flavor should be specified.

Response 3: This sentence explained the method and technical content. And the compounds have been specified.

 L:14-15:„ It was resulted that there were 7, 7 and 6 characteristic flavor compounds in popcorn, corn and lotus root flavor respectively.“ - I believe that it should be specifically stated which components these are.

Response 4: The characteristic flavor compounds of each type have been specified.

L:16-17: „ The 16 correlation between flavor volatiles and functional components was analyzed.“- I believe that the result of the correlation analysis should be stated.

Response 5: The result of the correlation analysis has been stated, and the correlation coefficient has been added.

 L:17-18: „ The resistant starch  content of lotus root flavor rice was relatively high.” - Please provide specific values.

Response 6: The values of the resistant starch content of lotus root flavor rice has been added.

L: 18-20: “Fat acidity of fragrant rice was highly correlated with the characteristic flavor compounds…” - Please state the value of the correlation coefficient.

Response 7: The correlation coefficient has been added.

The abstract lacks a conclusion.

Response 8: The abstract has been reorganized.

In general, I think that it is necessary to reformulate the abstract and specify the concrete results of the analysis. I find the abstract in this form to be very general/non-specific.

Response 9:  The abstract has been supplemented and reorganized.

Keywords: Keywords should complement and not repeat the title. The aim of keywords is to increase visibility of article in the databases, therefore more general words might be relevant, or words that describe the work, but are not included in the title. So keywords need to be changed (the ones in the title should not be repeated).

Response 10:  The keywords has been changed.

L: 24: „Rice is the main food crop in the world.“ - Rather, I would say - one of the main...

 Figure 1. Congratulations to the authors on a very clear and picturesque diagram :)

Response 11:  This sentence has been revised.

 L:264-265 „The resistant starch content of lotus root flavor rice was slightly higher than that of the corn flavor, and the popcorn flavor was the lowest.” - I think this is a general sentence. Please give specific values.

Response 12:  The specific values of the resistant starch contents have been added to this sentence.

 L:267-268 „The resistant starch content of different flavor type was different, and that of lotus  root flavor rice was relatively high.“ - Same comment as above. Please give specific values.

Response 13:  The specific values of the resistant starch contents have been added.

 L.262:   3.6. Correlation analysis between flavor volatiles and functional components - Please provide specific correlation coefficients/data for the relationships you state. You can display them in this part of the text or in a table.

Response 14:  The correlation coefficient of the correlation analysis has been added.

Reviewer 3 Report

This manuscript entitled “Characteristic flavor compounds and functional components of fragrant rice with different flavor types” investigated the various flavor types in fragrant. The topic is interesting and the manuscript has been written very well; however, the manuscript has several problems. 

  1. The keywords must be changed, popcorn flavor, corn flavor and lotus root flavor instead of flavor type; GC-MS; GC-O are recommended.
  2. Write the centigrade symbol in all the text (line 94, 95, 110, ….. )
  3. Lines 127 and 128 and throughout the text put the year of the references in brackets, for example: Zhou et al. (2022)
  4. Please check the word spacing throughout the manuscript.

Author Response

1. The keywords must be changed, popcorn flavor, corn flavor and lotus root flavor instead of flavor type; GC-MS; GC-O are recommended.

Response 1: The keywords has been changed.

2. Write the centigrade symbol in all the text (line 94, 95, 110, ….. )

Response 2: The centigrade symbol in all the text has been checked.

3. Lines 127 and 128 and throughout the text put the year of the references in brackets, for example: Zhou et al. (2022)

Response 3: The year of the references in brackets have been added.

4. Please check the word spacing throughout the manuscript.

Response 4: The word spacing throughout the manuscript has been checked.

Round 2

Reviewer 1 Report

I accept the manuscript.

Author Response

I accept the manuscript.

√ Thank you for your positive response.